# Development and Evaluation of a Full-Waveform Resistance Training Monitoring System Based on a Linear Position Transducer

**DOI:** 10.3390/s23052435

**Published:** 2023-02-22

**Authors:** Changda Lu, Kaiyu Zhang, Yixiong Cui, Yinsheng Tian, Siyao Wang, Jie Cao, Yanfei Shen

**Affiliations:** 1AI Sports Engineering Laboratory, School of Sports Engineering, Beijing Sport University, 48 Xinxi Road, Beijing 100084, China; 2Key Laboratory of Biomimetic Robots and Systems, Ministry of Education, School of Optics and Photonics, Beijing Institute of Technology, Beijing 100081, China

**Keywords:** linear position transducer, system design, full-waveform training monitoring system, resistance training, movement velocity

## Abstract

Recent advances in training monitoring are centered on the statistical indicators of the concentric phase of the movement. However, those studies lack consideration of the integrity of the movement. Moreover, training performance evaluation needs valid data on the movement. Thus, this study presents a full-waveform resistance training monitoring system (FRTMS) as a whole-movement-process monitoring solution to acquire and analyze the full-waveform data of resistance training. The FRTMS includes a portable data acquisition device and a data processing and visualization software platform. The data acquisition device monitors the barbell’s movement data. The software platform guides users through the acquisition of training parameters and provides feedback on the training result variables. To validate the FRTMS, we compared the simultaneous measurements of 30–90% 1RM of Smith squat lifts performed by 21 subjects with the FRTMS to similar measurements obtained with a previously validated three-dimensional motion capture system. Results showed that the FRTMS produced practically identical velocity outcomes, with a high Pearson’s correlation coefficient, intraclass correlation coefficient, and coefficient of multiple correlations and a low root mean square error. We also studied the applications of the FRTMS in practical training by comparing the training results of a six-week experimental intervention with velocity-based training (VBT) and percentage-based training (PBT). The current findings suggest that the proposed monitoring system can provide reliable data for refining future training monitoring and analysis.

## 1. Introduction

Resistance training (RT) is necessary for the development of muscle strength and is a fundamental component of many athletes’ periodized training programs [1,2]. Monitoring variables of RT allows coaches to improve their athletes’ training performance and optimize corresponding training plans. The RT’s effect depends on the operation of training variables such as the load, volume, and type of exercise [3,4,5].

Traditionally, the configuration of RT load is dependent on the percentage of the one-repetition maximum (1RM). Given the daily changes in 1RM, the actual and proposed loads generally do not match [3,6]. Recent studies have concentrated on the kinematic parameters represented by movement velocity to solve the problem of mismatched training loads [7,8]. Movement velocity could estimate the 1RM based on the velocity recorded against submaximal load based on load-velocity (L-V) relationships [9,10], prescribe the training loads and volumes accurately and objectively based on the magnitude of velocity loss [11,12], and increase motivation by providing real-time velocity feedback [13,14,15,16]. Furthermore, muscular and functional performance improves with small incremental changes in velocity relative to some reference loads in well-trained athletes [7,17,18,19]. Thus, it is essential to measure movement velocity with an accurate and reliable monitoring system.

In previous studies, the three-dimensional (3D) motion capture system was widely considered a “gold standard” instrument for measuring movement velocity [20,21,22,23]. However, the 3D motion capture system was not practical for daily training and test scenarios. Therefore, systems based on the inertial measurement unit (IMU) [24,25], camera [26], laser rangefinder [27], and linear position transducer (LPT) [28,29,30] were widely used in those scenarios because of their convenience and low cost [31,32,33]. Among the above systems, the LPT-based system was considered the most valid and reliable measurement system [29,34,35,36].

Previous validity and reliability research about these systems mostly concentrated on the concentric phase of the movement [29,31,37,38,39,40]. These studies usually monitored the statistical indicators represented by the mean or peak value of velocity during training [41], but these indicators may ignore the full waveform data of the whole movement [42]. Studies have shown that movements are characterized by a combination of concentric and eccentric muscle action in almost any type of sport [43]. Moreover, the eccentric phase of the training movements could optimize improvements to maximal muscular strength and power development and prevent sports injuries [44,45,46,47]. Menrad et al. used the MV of concentric and eccentric phases on Bland-Altman-diagrams and linear regression to test the accuracy of three systems (LPT, IMU, and PUSH.2.0) during three exercises (back squat, deadlift, and barbell rowing). LPT and IMU provided valid results determining MV in all three exercises, but data provided by PUSH 2.0 were not fully valid [48]. Concentric and eccentric phase studies used the statistical indicators of velocity to evaluate the validity of the system. Unfortunately, none of these studies focused on the validity of the full-waveform data during the movement.

To monitor the training process more comprehensively and provide valid data for later research on the refinement of training, this study has developed the full-waveform resistance training monitoring system (FRTMS) to acquire the full-waveform data and has verified its validity for measuring velocity during the whole movement. This study also applied the FRTMS to a real training scenario to explore its effects on training.

## 2. Materials and Methods

### 2.1. Full-Waveform Resistance Training Monitoring System

The monitoring system is designed and developed to fulfill the following three requirements:Measuring the displacement and velocity of the barbell during resistance training;Recording the training full-waveform data to quantify the training process and calculate the result variables to evaluate athletes’ performance;Providing real-time feedback on training parameters to coaches and athletes to help optimize the training process.

Based on these requirements, the FRTMS is composed of two parts: (i) a portable data acquisition device equipped with an optical incremental encoder and a hall encoder, which monitors the motion of the lift, and (ii) a software platform which remotely operates the device for data monitoring, calculation, and visualization.

The working process of the FRTMS is shown in Figure 1. The data acquisition device is placed under the fitness equipment (e.g., Smith machine and barbell) by four magnetic feet, while its strap is attached to the device’s tether with the barbell. During training, the linear position transducer (LPT) monitors the barbell’s movement and transmits the raw data to the software platform via Bluetooth, where they are processed and visualized to inform users about their actual performance in real time.

#### 2.1.1. Data Acquisition Device

The data acquisition device was designed to measure and calculate the position and velocity of the path of the lift and transmit the data to the software platform.

The maximum length and width of the device are 132 and 87 mm, respectively (Figure 2b). Owing to the 45# steel device frame, the total weight of the device (including electronics and sensors) is 1.15 kg. Thus, the device can be handily used for velocity-based training (VBT) in various application scenarios.

The housing includes: (i) a main control circuit with LEDS that show the most relevant real-time information about the system: 4 (in a row) representing the Bluetooth states, and 4 (in a column) indicating remaining battery capacity; (ii) a USB Type-C Connector for charging the battery; and (iii) a reset button (Figure 2a).

Compared to other LPTs, this device (Figure 2c) includes a Velcro strap and tether (for attaching to a bar or the pin of any pin-loaded weights machine) and magnetic feet for holding it in position. The device also has an added anti-wrap kit and a surface-mount slide guide roller, to reduce the probability of twining and knotting.

To measure the displacement profile of the lift, the data acquisition device is equipped with an optical encoder and a hall encoder.

A reflective surface mounts optical encoder, AEDR-8300 (Avago Technologies Limited, Singapore) was used. In cooperation with the code wheel, the encoding resolution of the encoder is 150 lines/inch with a pulse width error of 16°e and a phase error of 10°e at typical mounting alignment, which properly quantifies the displacement of the bar (threading the Velcro strap around) (Figure 3).

A mini 360° hall-effect angle encoder (SEEK FREE, Chengdu, China) was also used. The angle resolution of the encoder is 0.088°, which improves the accuracy of the displacement profile by measuring the angle of lift.

Beyond the previous two main encoders, the electronics of the data acquisition device include the following components (Figure 3a):

A Bluetooth 4.2 module E104-BT02 (Chengdu Ebyte Electronic Technology Limited, Chengdu, China) transfers the raw motion data to the host software platform.

Bluetooth LEDs (in a row) show the device and Bluetooth states (i.e., system mode, connection state, and transmission state).

Battery LEDs (in a column) show the remaining battery capacity.

A Battery Management System (BMS) controls a LiPo battery (3.7 V, 2800 mAh, which supplies the signal processing and encoder boards.

A microprocessor (STC16F40128K, 16-bit microcontroller at 24 MHz, STC micro, Guangdong, China) communicates with other components of the data acquisition device (Figure 3a). The processor’s counter counts pulses from the reflective optical encoder and the acquired pulse number is analyzed by the motion algorithm. Then, the processor immediately transmits the motion information (i.e., displacement, velocity, and acceleration) to the Bluetooth module 4.2. On the basis of the interface characters of different electronic components, the microprocessor adopts corresponding serial peripheral interfaces: SPI for the angle encoder, universal asynchronous receiver/transmitter UART for the Bluetooth module, and inter-integrated circuit interface I2C for the BMS. The firmware running in the microprocessor was programmed in C language. The raw angle and motion data are transmitted to the host software platform in real time (100 Hz).

For data transmission, the device communicates with the software platform via the user-defined protocol. The user-defined frame formats were divided into “request frame” and “response frame”. The formats of the request and the response frame are shown in Figure 4. “Instruction” and “Parameter” refer to the command to control the device. “Device Id” and “Battery level” refer to the status of the device. “Time” refers to the current boot time of the device. “Angle” refers to the strap’s current angle relative to the zero point of the device. “Position” refers to the strap’s current position relative to the zero point of the device. “Checksum” is added to the message frame to ensure data accuracy.

#### 2.1.2. Software Platform

To facilitate the use of the FRTMS, a software platform was developed using JavaScript-Html and Java language. The primary function of the software platform is to control the hardware device through the human-computer interaction interface, to acquire the data collected by the linear position sensor. Following processing and calculation, the device visualizes the result variables and action waveform and gives the user feedback to monitor their training. The software platform has two function modules: a data processing module and a visualization module.

The data processing module implements the filtering of the training data received from the sensors, using various algorithms, and calculates result variable values. Given that errors in the raw data are amplified in the metrics calculation, we must filter the raw data acquired from linear sensors to calculate the training variables and the full-waveform data accurately. The sampling frequency of the hardware device is 100 Hz. To eliminate the high-frequency noise, we use the “moving average” algorithm, which averages each data point with the two data points immediately preceding it.

We also applied a movement judgment algorithm to obtain valid training data. Using the instantaneous velocity at 0.05 m/s as a threshold to distinguish the beginning and end of a movement [49], a movement was considered to have begun when the instantaneous velocity was greater than 0.05 m/s and the duration of the movement was greater than 500 milliseconds. When the instantaneous velocity fell below 0.05 m/s, the movement was considered to have ended.

All training variables were determined using data from the beginning to the end of the movement. The training variables include Position, Mean Velocity, Max Velocity, Mean Power, Max Power, Relative Mean Power, and Relative Max Power.

Position (m) refers to the difference between the zero point and the current position (or the depth, if the device is fixed above).Mean Velocity (m/s) and Max Velocity (m/s) refer to the average and maximum values of instantaneous velocity in the concentric phase of a movement.Mean Power (W) and Max Power (W) refer to the average and maximum values of instantaneous power in the concentric phase of a movement. The first derivative of velocity is used to obtain acceleration, the mass of the training load, and body mass, and is combined with the choice of movement (since, because of the center of gravity, the value of work done by self-weight varies among different exercises).


(1)
$$P=\left(M_{1}+M_{2}\right)\times \left(g+a\right)\times V$$


Instantaneous power *P* applied to the system is calculated above, where *M*_1_ is the body mass of the users, *M*_2_ is the mass of the training load, *g* is the acceleration of gravity, *a* is the acceleration of the barbell, and *V* is the velocity of the barbell.Relative Mean Power (W/kg) and Relative Max Power (W/kg) refer to the average and maximum values of instantaneous power in the concentric phase of the movement, divided by the users’ body mass.

The Position (m) variable is used to provide the range of training movement (e.g., the depth of the squat and the height of the countermovement jump). Traditional training exercises usually use the mean variables (Mean Velocity and Mean Power) because the mean variables can appropriately reflect the athlete’s ability to lift the load throughout the concentric phase. For ballistic training (squat jump and bench press throw), peak variables (Max Velocity and Max Power) are more reliable for evaluation than velocity variables because of the difficulty velocity variables have in distinguishing flight thresholds. Relative variables (Relative Mean Power and Relative Max Power) are used to monitor relative strength training.

The visualization module provides the graphical display of instantaneous data and result variables. During training, the training variables and instantaneous data of the selected parameters are recorded and saved in the local database and then visualized by three different charts through the visualization interface (Figure 5a). A full-waveform graph displays the instantaneous data of each repetition, which can better reflect the details of changes throughout the entire training process. A monitoring variables chart shows the selected training variables, which helps the user to quickly see their training performance. Individual performance ranking charts rank the user’s training performance, motivating athletes to break through their personal bests as well as improve their performance within their team. The history training result variables and waveform data for each exercise are also visualized by line charts, bar charts, and sorting charts. Users can use history data to compare changes in training performance for a more comprehensive analysis and to support the development and adjustment of future training plans (Figure 5b).

### 2.2. Validity of the Full-Waveform Resistance Training Monitoring System

To assess the validity of the FRTMS, an experiment was designed and performed. The FRTMS data were validated by measuring velocity during the Smith squat exercise using the 3D motion capture system. In this experiment, subjects participated in two testing sessions: baseline and formal testing sessions.

#### 2.2.1. Subjects

A total of 13 men and 8 women volunteered for this study (mean ± SD: 23.5 ± 1.5 years; 1.75 ± 0.08 m; 69.9 ± 11.0 kg; back squat 1RM: 95.5 ± 27.1 kg). All subjects were familiar with the back squat prior to the commencement of the study. They had no health issues, physical limitations, or musculoskeletal injuries that could affect their performance in the test.

Before the beginning of the study, all subjects were informed of the procedures and signed a written informed consent form. The study protocol adhered to the principles of the Helsinki Declaration and was approved by the institutional review board.

#### 2.2.2. Equipment Setup

The OptiTrack system is a 10-camera 3D motion capture system employed as the “gold standard” reference system. Before the test, the entire calibration of the equipment was completed in accordance with the OptiTrack user guide. Cameras also tracked the marker (14 mm in diameter) placed at the end of the barbell, allowing bar movement to be measured (Figure 6).The FRTMS is a training monitoring system in which the data acquisition device is attached to the Smith machine’s side and the tether is bound to the rack’s barbell. Before each squat, the tether was checked by the researcher to ensure its perpendicularity (Figure 6).

#### 2.2.3. Experimental Procedure

During the baseline session, each subject’s body mass and height were measured. Self-reported 1RMs and relative loads for the Smith squat were used to determine training loads. The subjects were asked to refrain from vigorous lower body exercise 48 h before the formal examination. Since studies had shown that the protective equipment could improve movement velocity [50], subjects were prohibited from wearing any protective equipment (wrist guard, belt, or knee strap) during the examination to ensure uniformity and accuracy of measurements.

During the formal testing session, subjects performed a 15-minute general warm-up consisting of jogging, dynamic stretching, and joint mobilization exercises for the entire body. Subjects performed the Smith squat with five relative loads (three repetitions at 30%, 45%, 60%, and 75% of 1RM and one repetition at 90% of 1RM) that were implemented in incremental order. Before each set began, subjects were instructed to perform the concentric portion of each repetition as explosively as possible. They rested for two minutes between each set and a maximum of ten seconds between reps. After a 1–2 s static pause between each squat, a researcher would give the command “Start!” A squat was regarded as successful when the greater trochanter was below the lateral epicondyle of the knee at the lowest point of descent and the participant could fully extend their lower limbs on an ascent. Moreover, subjects followed the high-bar back technique (barbells were placed above the subject’s trapezius muscle) and choose their own grip width during squats.

#### 2.2.4. Data Processing and Analysis

The OptiTrack system and the FRTMS simultaneously collected kinematic data at a rate of 100 Hz. To ensure synchronization of the collected data, subjects were instructed to wait a few seconds before beginning each squat, and the highest point after the barbell was removed from the rack was set to zero position.

For data acquisition and variable calculation, the OptiTrack system employs Motive 2.1 software (OptiTrack system’s unified software platform) while the FRTMS employs the proposed software platform (which records the full-waveform position data). The data of the OptiTrack system were filtered by a 6 Hz, low-pass, fourth-order Butterworth filter to remove any high-frequency noise. For a more realistic assessment of the accuracy of the device’s raw data, the FRTMS measurements were used unfiltered. All raw data were exported to CSV format and subsequently processed using Python (version 3.9).

As a note, the velocity data were calculated using raw position data, according to two different methods by the two devices. To align the time axes of the two systems, the data sets were convolved to determine the frame with the highest correlation (number of interrelationships) between the data, corresponding to each action. This frame was used as the basis for translating the time axis. Once the time axes were aligned, the data corresponding to the same time range could then be compared.

#### 2.2.5. Statistical Analysis

Validity analyses included the calculation of a set of statistics aimed at providing information about the accuracy of measurement incurred when using the technologies under study. In this study, overall validity was determined using data from each squat performed by 21 subjects whose data were available from both the OptiTrack system and the FRTMS. All statistical indications were calculated using Python (version 3.9). To determine the magnitude of errors at particular load ranges, statistical data and full-waveform data under each relative load (30%, 45%, 60%, 75%, and 90% of 1RM) were calculated separately. Furthermore, to compare the magnitude of errors at different squat phases, statistical data were calculated for concentric and eccentric phases. The statistical analyses conducted included Pearson’s correlation coefficient (r), ICC (two-way random, absolute agreement), and root mean square error (RMSE). The thresholds of ICCs were classified by the following criteria: perfect (ICC = 0.91–1.00), good (ICC = 0.76–0.90), moderate (ICC = 0.51–0.75), and poor (ICC = 0.00–0.50) [51]. Person’s correlation coefficient was classified using the following criteria: perfect (r = 0.91–1.00), very high (r = 0.71–0.90), and high (r = 0.50–0.70) [49]. The equations of all linear regressions and the coefficients of determination (R²) were drawn in Figure 7, Figure 8 and Figure 9 (where y = mx+ b; m is the relative deviation and b is the bias).

Furthermore, the coefficient of multiple correlation (CMC) was employed to evaluate the level of concordance between the full waveform data acquired by the two systems. Given that this metric was generally used to evaluate the similarity of movement cycle waveforms, a squat process was considered one movement cycle in this study. Therefore, the two systems’ waveform data were input into the formula to obtain the CMC value, rated as perfect (CMC = 0.95–1.00), very good (0.85–0.94), good (CMC = 0.75–0.84), moderate (CMC = 0.40–0.74), or poor (CMC = 0.00–0.39) [52].

## 3. Results

Validation studies have reported strong correlations between the FRTMS and 3D motion capture systems for a range of activities, including correlations in concentric mean velocity (r = 0.997–0.998, ICC = 0.998–1.000, and RMSE = 0.005–0.010) (Table 1), eccentric mean velocity (r = 0.997–0.999, ICC = 0.999–1.000, and RMSE = 0.005–0.008) (Table 2), and full-waveform velocity (r = 0.998–0.999, ICC = 0.999–1.000, RMSE = 0.017–0.028, and CMC = 0.999–1.000) (Table 3).

Figure 7, Figure 8 and Figure 9 show the linear regressions of the average velocities and full-waveform velocities of the FRTMS versus the reference system for Smith squats during different relevant loads and phases. A strong correlation was also shown in the regression analysis including concentric mean velocity (R^2^ = 0.9981–0.9997), eccentric mean velocity (R^2^ = 0.9988–0.9999), and full-waveform velocity (R^2^ = 0.9969–0.9984). Moreover, the bias b of all data for all loads and phases is small (b < 0.0029 m/s).

Table 4 shows the mean value ± standard deviation of the mean velocities and full-waveform velocities of the FRTMS and the OptiTrack system in different loads and different phases. The mean values of the FRTMS were lower than those of the OptiTrack system.

## 4. Discussion

In this study, we proposed and validated the FRTMS as a whole-movement-process monitoring solution to acquire and analyze the full-waveform data of resistance training.

Though previous studies have verified the validity of different systems from different relative loads by measuring various statistical metrics, none of these studies focused on the validity of the data during the whole movement. Ferro et al. used the concentric phase’s Peak Velocity (PV) to evaluate the typical error (TE) and used the ICCs to verify that the LPT, IMU, and force platform could provide reliable measurements during the countermovement jump exercise [53]. McGrath et al. used the concentric phase’s mean velocity (MV) for linear regression and evaluated the intraclass correlation coefficients (ICC) to verify that the LPT could provide valid data during the bench press [54]. Thompson et al. used the concentric phase’s MV and PV to evaluate the TEs and coefficient of variations (CVs), to verify that the LPT provided the most valid and reliable measurements for the back squat and power clean, followed by the camera-based system and IMU for the back squat [40].

Compared with these studies, our study supplemented the CMC as a validity indicator to evaluate the level of concordance of the full waveform data acquired by various systems.

The results for the FRTMS corresponded to the findings of other studies on LPT systems, and the FRTMS had a strong correlation with the OptiTrack system.

According to Askow et al., GymAware was a valid system for measuring MV during the back squat exercise. They also used a 3D motion capture system (Qualysis, Sweden) as the “gold standard” reference system. All participants performed squats at two relative loads (75% and 90% of 1RM). Their results showed that LPT overestimated the MV by 0.03 m/s [20].

McGrath et al. compared the validity of Tendo and PUSH 1.0 in the squat with a 3D motion capture system (Santa Rosa, California). Ten subjects performed bench presses at two relative loads (40% and 80% of 1RM). The results showed that the LPT overestimated mean concentric velocity (MCV) at higher loads (80% of 1RM), but underestimated the velocity at lower loads (40% of 1RM) [54].

The explanation for why the FRTMS underestimated the velocity may be associated with the different resolution accuracy for the displacement of the FRTMS and OptiTrack systems. The LPT has a slightly lower displacement resolution than the motion capture system. Thus, the OptiTrack system can capture small displacement and velocity changes before the start and after the end of the motion, whereas the FRTMS cannot. Thus, the OptiTrack system will incorporate these small velocity values into the MV calculation, making its MV larger than that of the FRTMS.

To further test the system’s application in practical training, the FRTMS was also used in an intervention study that compared the effects of a 10% velocity loss threshold VBT intervention and a traditional percentage-based training (PBT) intervention on lower limbs’ explosive power (Figure 10). The study consisted of three 1RM tests (pre-test, mid-test, and post-test) and a six-week, twice-weekly back squat intervention. Before the intervention study, all participants who met the criteria for the experiment were informed of the procedures and signed a written informed consent form. The study protocol adhered to the principles of the Helsinki Declaration and was approved by the institutional review board. During the 1RM test, the LPT of the FRTMS was attached to the barbell to monitor the MCV of each squat.

Preventing sports injury: the participants avoided attempting heavier loads when their MCV was below the squat minimum velocity threshold (0.3 m/s) [55].Monitoring movement quality: the FRTMS also monitored the depth of each squat, ensuring that all repetitions were performed at a qualified depth.

During the intervention, the system monitored the MCV, position, and instantaneous velocity of the VBT group.

Providing velocity feedback: Real-time velocity feedback improved the participants’ training motivation and assisted the coach with performing the intervention.Adjusting training load: if today’s reference velocity (the MCV of the first squat during 80% of 1RM) was 0.06 m/s lower or higher than the corresponding velocity at the 1RM test, the training load needs to be adjusted by ±5% of the tested 1RM [12].

During data processing, the full-waveform velocity of three 1RM tests, as well as the MCV and the maximum concentric velocity of each squat, during the six-week intervention provided by the monitoring system were used in the statistical process.

Monitoring the enhancement of movement: by comparing the duration of the sticking region and the depth of squats before and after the intervention, the full-waveform velocity can indicate the enhancement of the squat action.Monitoring the detailed improvement of the training performance: comparing the change in the maximal concentric velocity (MCV) and the time to peak velocity during the intervention can reflect detailed improvement of the training performance. For example, after a training intervention, the 1RM value of the athletes did not change. However, their time to peak velocity and the duration of the sticking region each decreased by 50 milliseconds, which for athletes could be of major benefit to the explosive power of the lower limbs [42].

This study tested the validity of the FRTMS during squats performed on the Smith Machine. However, the daily training also included other multi-directional free weight exercises different from the limited movement patterns used in research. Additionally, the subjects of this study did not represent all users of this system. Based on the above limitations, future studies would recruit more subjects to examine the validity of the FRTMS during more free weight exercises, such as snatch, power clean, and jerk.

## 5. Conclusions

The current study introduced a recently developed FRTMS and validated its accuracy when measuring the full-waveform data and kinetic and kinesiological metrics during the whole movement process of resistance exercises. Proof of the validity of the monitoring system was established by comparing the velocity measurements taken simultaneously by the FRTMS versus a gold-standard 3D motion capture system during 30–90% 1RM of Smith squat lifts. Moreover, a case study was described to expound on the system’s application in practical training and experiments. In summary, the proposed monitoring system can provide reliable data for future refinement of training monitoring and analysis.

## Figures and Tables

**Figure 1 sensors-23-02435-f001:**
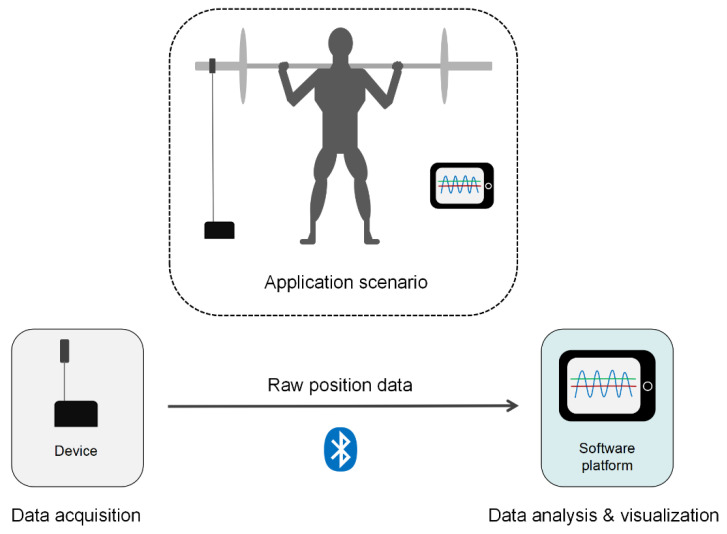
Overview of the full-waveform resistance training monitoring system (FRTMS). The device communicates with the software platform via Bluetooth.

**Figure 2 sensors-23-02435-f002:**
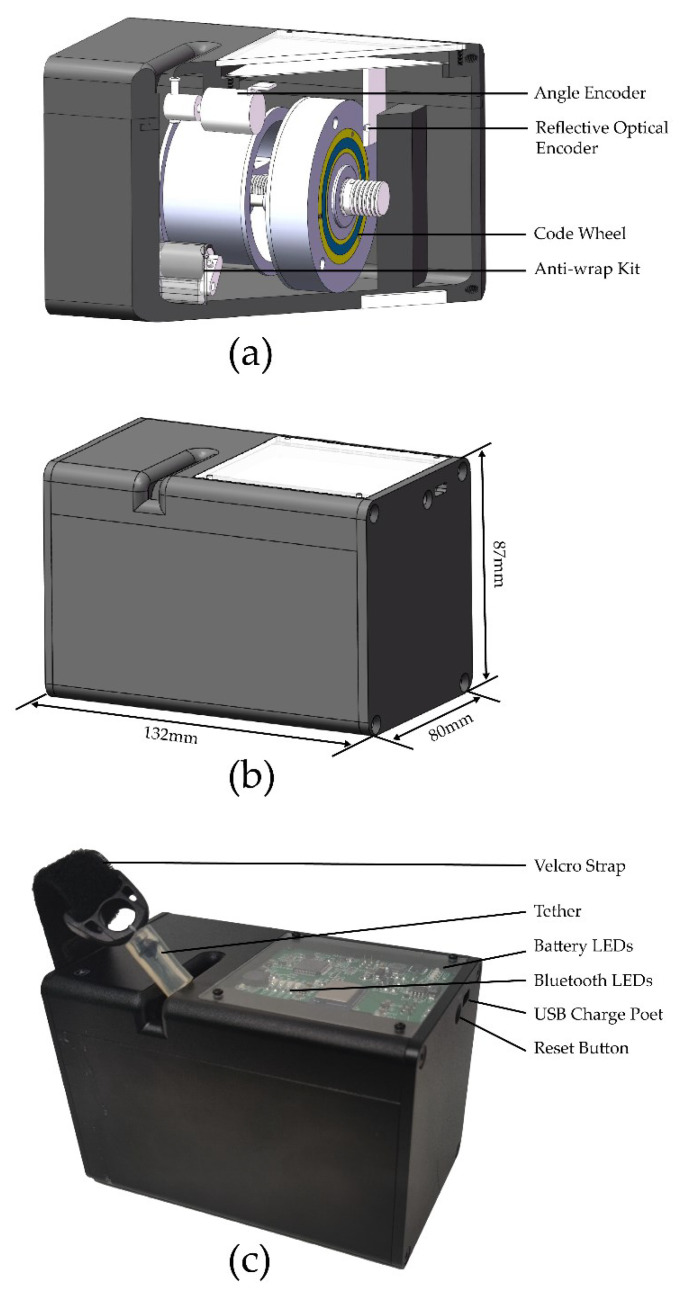
Data acquisition device overview: (**a**) 3D CAD model and components; (**b**) assembly drawing; (**c**) physical prototype.

**Figure 3 sensors-23-02435-f003:**
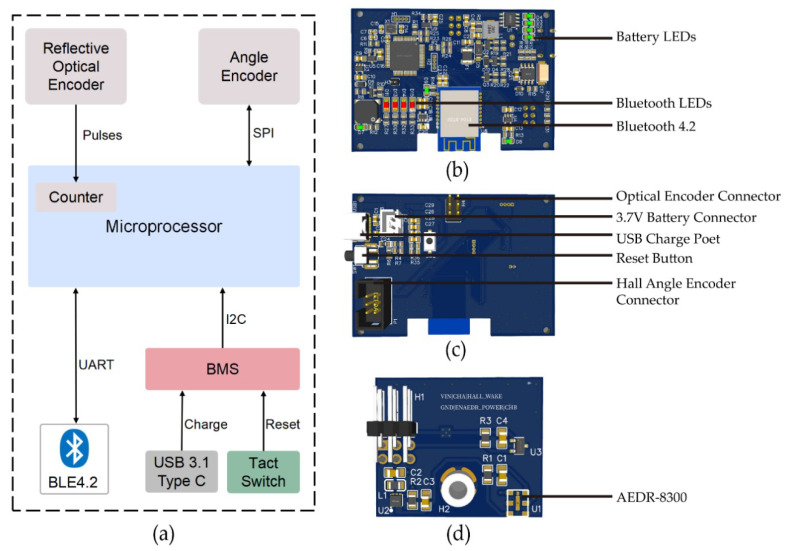
Electronic connections: (**a**) diagrammatic drawing; (**b**) the top layer of the main control circuit board; (**c**) the bottom layer of the main control circuit board; (**d**) the layout of the mini circuit board for displacement measurement.

**Figure 4 sensors-23-02435-f004:**
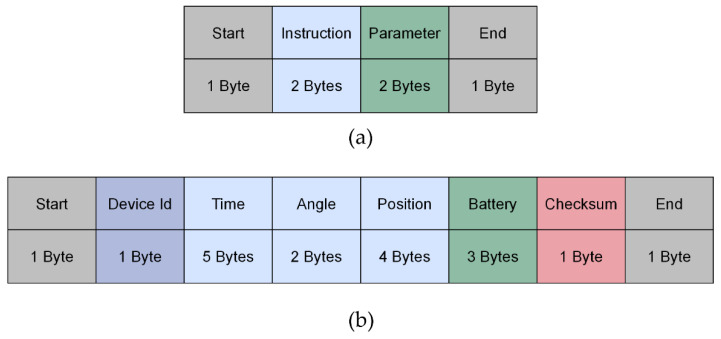
The format of the message frame: (**a**) request frame; (**b**) response frame.

**Figure 5 sensors-23-02435-f005:**
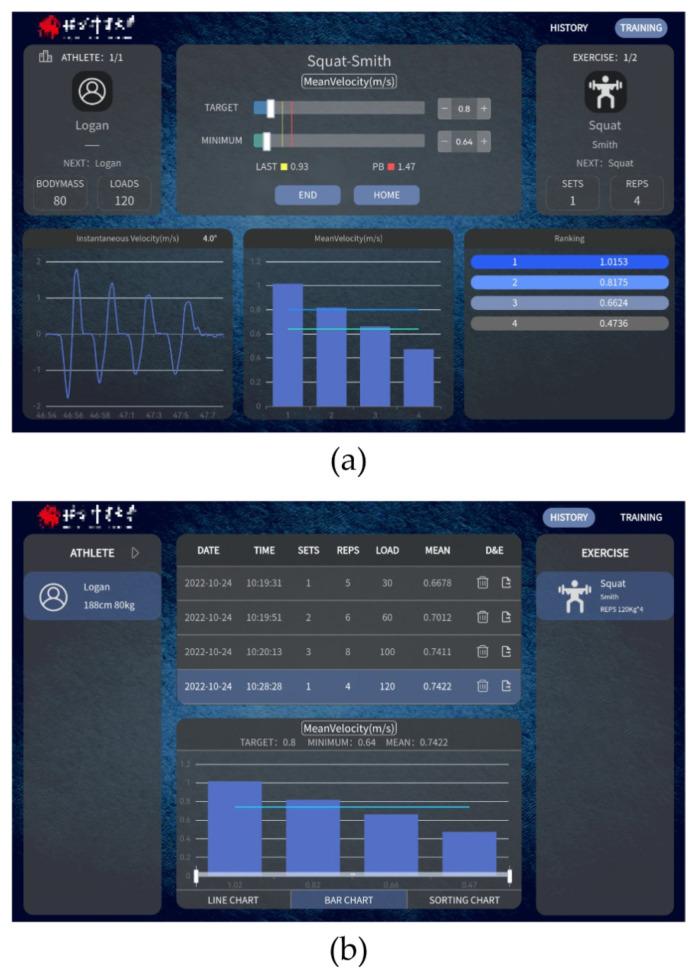
Visualization interface of the software: (**a**) training data interface; (**b**) history data interface.

**Figure 6 sensors-23-02435-f006:**
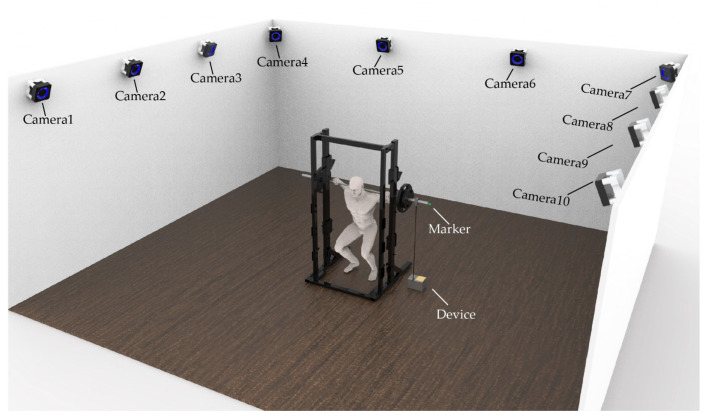
Distribution of the monitoring systems in the test.

**Figure 7 sensors-23-02435-f007:**
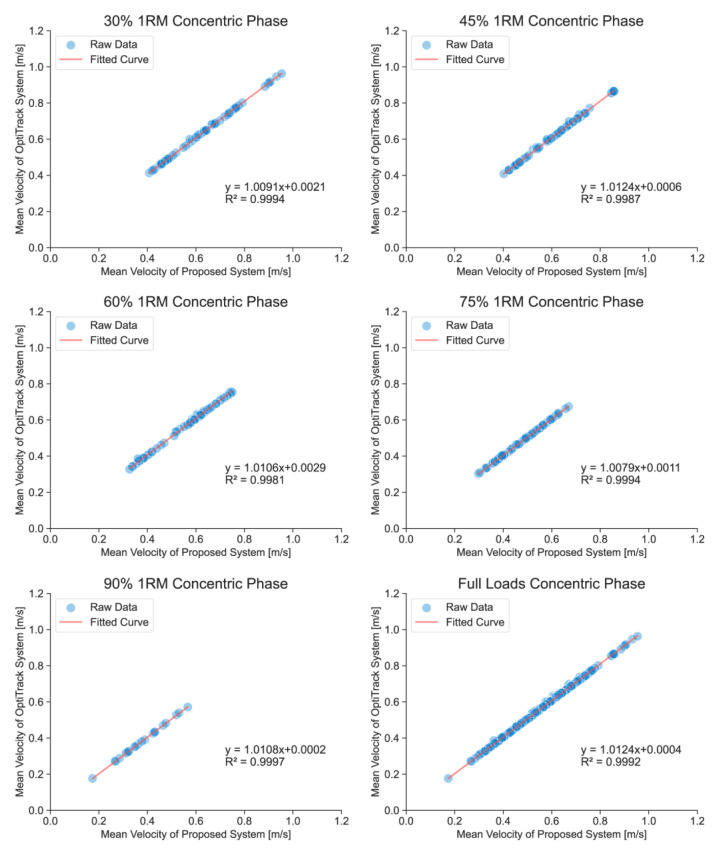
Linear regressions of mean velocities in the concentric phase under different loads, measured by the FRTMS versus the OptiTrack system.

**Figure 8 sensors-23-02435-f008:**
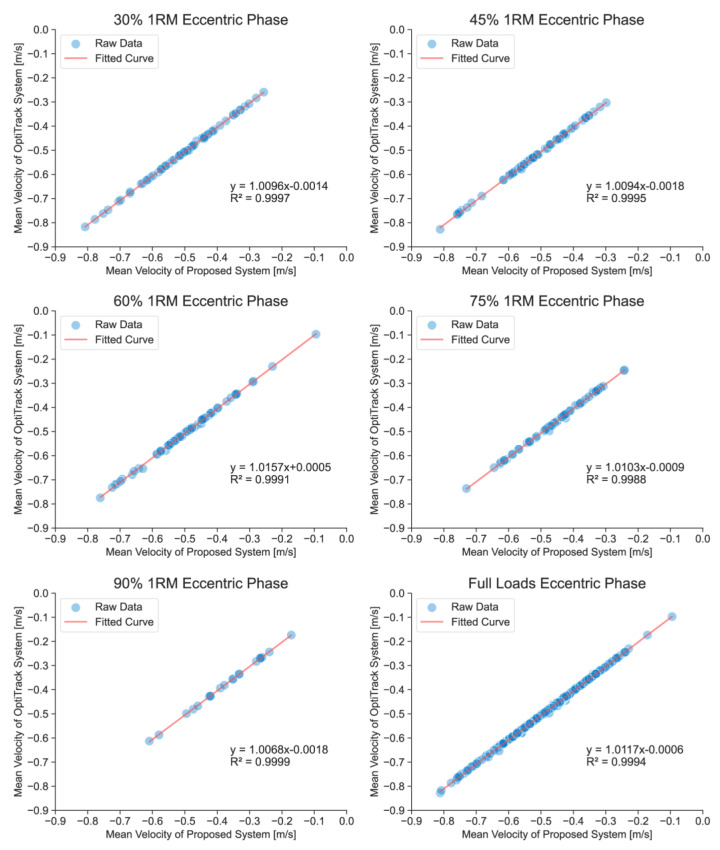
Linear regressions of mean velocities in the eccentric phase under different loads, measured by the FRTMS versus the OptiTrack system. The Full Loads graph is the regression of all relative loads’ average velocities. The positivity or negativity of the velocities represents the direction of movement: negative velocity means downward movement, while positive velocity means upward movement.

**Figure 9 sensors-23-02435-f009:**
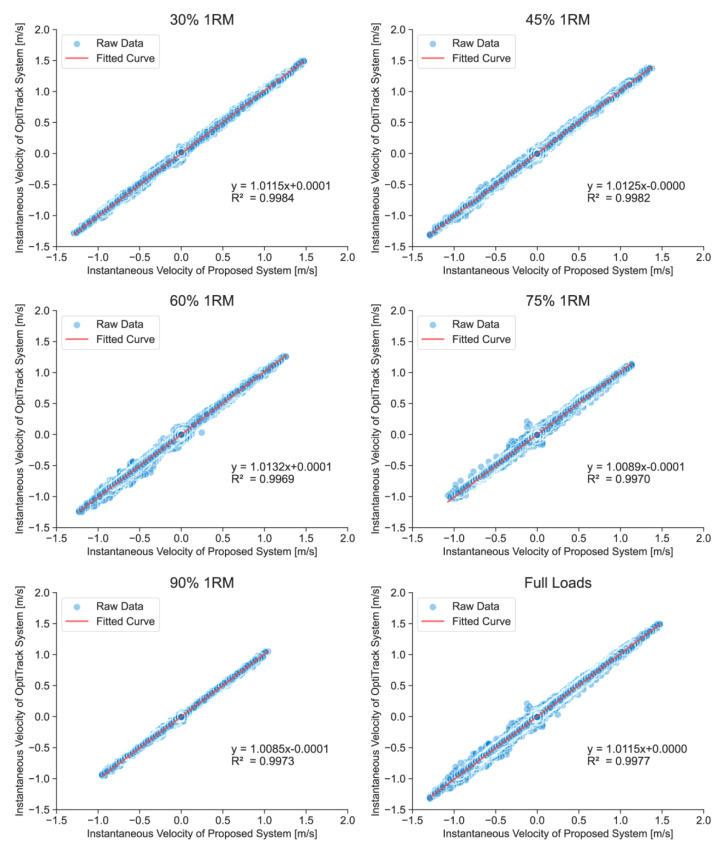
Linear regressions of the instantaneous velocities under different loads, measured by the FRTMS versus the OptiTrack system. The Full Loads graph is the regression of all relative loads’ instantaneous velocities. The positivity or negativity of the velocities represents the direction of movement: negative velocity means downward movement, while positive velocity means upward movement.

**Figure 10 sensors-23-02435-f010:**
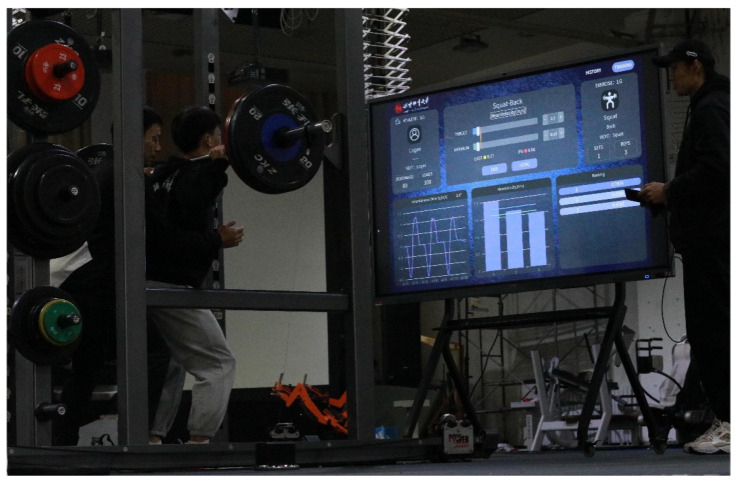
The FRTMS being used in squat training.

**Table 1 sensors-23-02435-t001:** Pearson’s correlation coefficient (r), intraclass correlation coefficient (ICC), and root mean square error (RMSE) of concentric mean velocity measured by the full-waveform resistance training monitoring system (FRTMS) and the OptiTrack system during different loads.

Loads (%RM)	r	ICC (2,1)	RMSE
30%	0.997	0.999	0.009
45%	0.998	0.999	0.010
60%	0.997	0.998	0.010
75%	0.997	0.999	0.005
90%	0.998	1.000	0.005
All	0.998	0.998	0.009

**Table 2 sensors-23-02435-t002:** Pearson’s correlation coefficient (r), ICC, and RMSE of eccentric mean velocity measured by the FRTMS and the OptiTrack system during different loads.

Loads (%RM)	r	ICC (2,1)	RMSE
30%	0.998	0.999	0.007
45%	0.999	0.999	0.007
60%	0.998	0.999	0.008
75%	0.998	0.999	0.007
90%	0.997	1.000	0.005
All	0.997	0.999	0.007

**Table 3 sensors-23-02435-t003:** Pearson’s correlation coefficient®, ICC, RMSE, and coefficient of multiple correlation (CMC) of full-waveform velocity measured by the FRTMS and the OptiTrack system during different loads.

Loads (%RM)	r	ICC(2,1)	RMSE	CMC
30%	0.999	1.000	0.021	1.000
45%	0.999	0.999	0.023	0.999
60%	0.998	0.999	0.028	0.999
75%	0.998	0.999	0.024	0.999
90%	0.999	0.999	0.017	0.999
All	0.999	0.999	0.023	0.999

**Table 4 sensors-23-02435-t004:** Mean values of concentric mean velocity (CON), eccentric mean velocity (ECC), and full-waveform velocity between the FRTMS and the OptiTrack system (OptiTrack) under different loads [means ± SD].

Loads (%RM)	CON		ECC		Full-Waveform	
	FRTMS (m/s)	OptiTrack (m/s)	FRTMS (m/s)	OptiTrack (m/s)	FRTMS (m/s)	OptiTrack (m/s)
30%	0.629 ± 0.146	0.637 ± 0.147	0.504 ± 0.126	0.510 ± 0.127	0.384 ± 0.340	0.391 ± 0.342
45%	0.606 ± 0.126	0.614 ± 0.128	0.520 ± 0.121	0.528 ± 0.122	0.392 ± 0.331	0.399 ± 0.333
60%	0.554 ± 0.123	0.563 ± 0.124	0.494 ± 0.124	0.501 ± 0.126	0.367 ± 0.309	0.375 ± 0.311
75%	0.476 ± 0.096	0.481 ± 0.097	0.451 ± 0.104	0.457 ± 0.105	0.329 ± 0.270	0.335 ± 0.270
90%	0.378 ± 0.096	0.382 ± 0.097	0.370 ± 0.111	0.374 ± 0.111	0.218 ± 0.225	0.223 ± 0.224

## Data Availability

Not applicable.

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
