# Peer review of "Development and Evaluation of a Full-Waveform Resistance Training Monitoring System Based on a Linear Position Transducer"

_sensors, 2023, doi:10.3390/s23052435_

Round 1

Reviewer 1 Report

Your study idea is interesting, my recommendations are the following:

I recommend the introduction to keywords and a- speed of movement.

In the last sentence of section 1, use the previously mentioned abbreviations.

I recommend that you specifically mention what the preparation included during the 6 weeks of training.

Extension of the Discussions section with relevant previous studies to support the results of this study.

I recommend that you mention the limitations of this study.

I congratulate you for this idea and the clear way of presenting the specific aspects.

Author Response

Response to Reviewer 1 Comments

On behalf of my co-authors, thank you very much for giving us an opportunity to revise our manuscript, we appreciate reviewers very much for your constructive and positive comments and suggestions which helped us improve our manuscript entitled “Development and Evaluation of a Full-waveform Resistance Training Monitoring System Based on a Linear Position Transducer”. (Manuscript ID: sensors-2156481).

We have studied reviewer’s comments carefully and have made revision which marked in red font in the copy of the revised manuscript. We have tried our best to revise our manuscript according to the comments from reviewers.

Thank you and best regards.

Yours sincerely,

Changda Lu

E-mail: luchangda@bsu.edu.cn

Reviewers' comments:

Your study idea is interesting, my recommendations are the following:

I recommend the introduction to keywords and a- speed of movement.

In the last sentence of section 1, use the previously mentioned abbreviations.

I recommend that you specifically mention what the preparation included during the 6 weeks of training.

Extension of the Discussions section with relevant previous studies to support the results of this study.

I recommend that you mention the limitations of this study.

I congratulate you for this idea and the clear way of presenting the specific aspects.

To reviewer 1

We thank reviewer 1 for the critical comments and helpful suggestions. We have taken all these comments and suggestions into account, and they have improved our manuscript considerably.

Point 1: I recommend the introduction to keywords and a- speed of movement.

Response 1: Thank you very much for your suggestions. According to your suggestions, we added Movement Velocity to the keywords.

Point 2: In the last sentence of section 1, use the previously mentioned abbreviations.

Response 2: Thank you very much for your reminder. After your reminder, we used the previously mentioned abbreviations in the sentence of section 1.

Revised details on manuscript:

The revised details of the manuscript are in chapter 1:

….

To monitor the training process more comprehensively and provide valid data for later research on the refinement of training, this study has developed the full-waveform resistance training monitoring system (FRTMS) to acquire the full-waveform data and verified its validity of measuring velocity during the whole movement. Additionally, this study also applied the FRTMS to a real training scenario to explore its effects on training.

….

Point 3: I recommend that you specifically mention what the preparation included during the 6 weeks of training.

Response 3: Thank you very much for your suggestions. According to your suggestions, we have added the preparation contents to the 6 weeks application study.

Revised details on manuscript:

The revised details of the manuscript are in chapter 4:

….

Before the intervention study, all participants who met the criteria of the experiment were informed of the procedures and signed a written informed consent form. The study protocol adhered to the principles of the Helsinki Declaration and was approved by the institutional review board.

….

Point 4: Extension of the Discussions section with relevant previous studies to support the results of this study.

Response 4: Thank you very much for your suggestions. According to your suggestions, we have compared the method of the validation with other three studies.

Revised details on manuscript:

The revised details of the manuscript are in chapter 4:

….

In this study, we proposed and validated the FRTMS as a whole movement process monitoring solution to acquire and analyze the full-waveform data of resistance training.

Despite the previous studies verifying the validity of different systems from different relative loads by measuring various statistical metrics, none of these studies focused on the validity of the data during the whole movement. Ferro et al. used the concentric phase’s Peak Velocity (PV) for evaluated the typical error (TE) and the ICCs to verify that the LPT, IMU and force platform could provide reliable measurements during the countermovement jump exercise[53]. McGrath et al. used the concentric phase’s mean velocity (MV) for linear regression and evaluated the intraclass correlation coefficients (ICC) to verify that the LPT could provide valid data during the bench press[54]. Thompson et al. used the concentric phase’s MV and PV for evaluated the TEs and coefficient of variations (CVs) to verify that the LPT provided the most valid and reliable measurements for the back squat and power clean, followed by camera-based system and IMU for the back squat[40].

Compared with these studies, our study supplemented the CMC as a validity indicator to evaluate the level of concordance of the full waveform data acquired by various systems.

….

Point 5: I recommend that you mention the limitations of this study.

Response 5: Thank you very much for your comments. According to your comments, we have added limitations of this study to the chapter 4 Discussion.

Revised details on manuscript:

The revised details of the manuscript are in chapter 4:

…..

This study tested the validity of FRTMS during squats performed on the Smith Machine. However, the daily training also included other multi-directional free weight exercises different from the limited movement patterns used in research. Additionally, the subjects of this study did not represent all users of this system. Based on the above limitations, future studies would recruit more subjects to examine the validity of FRTMS during more free weight exericises, such as snatch, power clean and jerk.

…..

Reviewer 2 Report

This study was conducted by unique system for assessing resistance training performance. This study will contribute to better coaching or training for coaches or athletes.  However, unfortunately, this paper is not written by proper research manuscript sections, that is, introduction, materials and methods, results, discussion and conclusions. After solving the basic problem severely, this paper should be resubmitted soon.

In addition, I comment this paper a bit.

The introduction section is written redundantly. The explanation for IMU-, camera-, laser-, and LPT-based systems is not needed.

4.3. Data processing Module: Change body weight to body mass because the authors are using kg here.

Author Response

Response to Reviewer 2 Comments

On behalf of my co-authors, thank you very much for giving us an opportunity to revise our manuscript, we appreciate reviewers very much for your constructive and positive comments and suggestions which helped us improve our manuscript entitled “Development and Evaluation of a Full-waveform Resistance Training Monitoring System Based on a Linear Position Transducer”. (Manuscript ID: sensors-2156481).

We have studied reviewer’s comments carefully and have made revision which marked in red font in the copy of the revised manuscript. We have tried our best to revise our manuscript according to the comments from reviewers.

Thank you and best regards.

Yours sincerely,

Changda Lu

E-mail: luchangda@bsu.edu.cn

Reviewers' comments:

This study was conducted by unique system for assessing resistance training performance. This study will contribute to better coaching or training for coaches or athletes.  However, unfortunately, this paper is not written by proper research manuscript sections, that is, introduction, materials and methods, results, discussion and conclusions. After solving the basic problem severely, this paper should be resubmitted soon.

In addition, I comment this paper a bit.

The introduction section is written redundantly. The explanation for IMU-, camera-, laser-, and LPT-based systems is not needed.

4.3. Data processing Module: Change body weight to body mass because the authors are using kg here.

To reviewer 2

We thank reviewer 2 for the critical comments and helpful suggestions. We have taken all these comments and suggestions into account, and they have improved our manuscript considerably.

Point 1: This paper is not written by proper research manuscript sections, that is, introduction, materials and methods, results, discussion and conclusions. After solving the basic problem severely, this paper should be resubmitted soon.

Response 1: Thank you very much for your suggestion. With your advice, we have rewritten the article in the format of “introduction, materials and methods, results, discussion and conclusions” according to the journal's guidelines.

Revised details on manuscript:

The structure of the revised manuscript is as follows:

…..

  1. Introduction
  2. Materials and Methods

       2.1 Full-waveform Resistance Training Monitoring System

              2.1.1 Data Acquisition Device

              2.1.2 Software Platform

2.2 Validity of the Full-waveform Resistance Training Monitoring System

2.2.1 Subjects

2.2.2 Equipment Setup

2.2.3 Experimental Procedure

2.2.4 Statistical Analysis

2.2.5 Data Processing and Analysis

  1. Results
  2. Discussion
  3. Conclusion

…..

Point 2: The introduction section is written redundantly. The explanation for IMU-, camera-, laser-, and LPT-based systems is not needed.

Response 2: Thank you very much for your suggestions. According to your suggestions, we removed the explanation for IMU-, camera-, laser-, and LPT-based systems to simplify the Introduction. Additionally, we have kept only the most important examples of the Introduction and moved others to Discussion for detailed discussion.

Revised details on manuscript:

The revised introduction section is as follows:

…..

Resistance training (RT) is necessary for the development of muscle strength and is a fundamental component of many athletes' periodized training programs[1,2]. Monitoring variables of RT allows coaches to improve their athletes' training performance and optimize corresponding training plans. The RT's effect depends on the operation of the training variables like the load, volume, and type of exercises[3,4,5].

Traditionally, the configuration of the RT’s load is depended on the percentage of one‐repetition maximum (1RM). Given the daily changes in 1RM, the actual and proposed loads make a mismatch[3,6]. Recent studies have concentrated on the kinematic parameters represented by movement velocity to solve the problem of mismatched training loads [7,8]. The movement velocity could estimate the 1RM from the velocity recorded against submaximal load based on load-velocity (L-V) relationships[9,10], prescribe the training loads and volumes accurately and objectively based on the magnitude of velocity loss [11,12], and increase motivation by providing real-time velocity feedback[13,14,15,16]. Furthermore, the muscular and functional performance improved with the small incremental changes in the velocity relative to some reference loads in well-trained athletes[7,17,18,19]. Thus, it is essential to measure movement velocity with an accurate and reliable monitoring system.

In previous studies, the three-dimensional (3D) motion capture system was widely considered a “gold standard” instrument for measuring movement velocity.[20,21,22,23].   However, the 3D motion capture system was not practical for daily training and test scenarios. Therefore, the systems based on the inertial measurement unit (IMU)[24,25], camera[26], laser rangefinder[27] and linear position transducer(LPT)[28,29,30] werewidely used in those scenarios because of their convenience and low cost[31,32,33]. Among the above systems, the LPT-based system was considered as the most valid and reliable measurement system [29,34,35,36].

Previous validity and reliability researches about these systems mostly concentrated on the concentric phase of the movement[29,31,37,38,39,40]. These studies usually monitored the statistical indicators represented by the mean or peak value of velocity during training[41], but these indicators for the whole movement may ignore the full waveform data of the movement[42]. Studies have shown that the movements were characterized by a combination of concentric and eccentric muscle action in almost any type of sport[43]. Moreover, the eccentric phase of the training movements could optimize improvements to maximal muscular strength, power development and prevent sports injuries[44,45,46,47]. Menrad et al. used the MV of concentric and eccentric phases on Bland-Altman-diagrams and linear regression to test the accuracy of the three systems during three exercises. LPT and IMU provided valid results determining MV in back squat, deadlift and barbell rowing, but PUSH 2.0 provided not fully valid data[48]. Concentric and eccentric phase studies used the statistical indicators of the velocity to evaluate the validity of the system. Unfortunately, none of these studies focused on the validity of the full-waveform data during the movement.

To monitor the training process more comprehensively and provide valid data for later research on the refinement of training, this study has developed the full-waveform resistance training monitoring system (FRTMS) to acquire the full-waveform data and verified its validity of measuring velocity during the whole movement. Additionally, this study also applied the FRTMS to a real training scenario to explore its effects on training.

…...

Point 3: 4.3. Data processing Module: Change body weight to body mass because the authors are using kg here.

Response 3: Thank you very much for your careful comment. We feel sorry for this mistake. We have changed body weight to body mass.

Revised details on manuscript:

The revised details of the manuscript are in 2.1.2 Software Platform:

…..

Relative Mean Power (W/kg) and Relative Max Power (W/kg) refer to the average and maximum values of instantaneous power in the concentric phase of the movement divided by the users’ body mass.

…..

Reviewer 3 Report

Dear Authors

The presented concept of a new measurement method is an interesting supplement to the existing solutions and is worth considering.

However, the text is very long and quite hard to read. There are too many goals in one article. I would suggest considering whether it would not be worth dividing this text into two or even three reports. The first topic is a review of existing studies and a comparison of methods. The second topic is the presentation of the new device with technical details. And the third topic is the use of the device in pilot studies and comparing the results with other methods. This would greatly simplify the message and the topics would be clearer and easier for the reader.

However, if you wish to publish it in this form, I suggest the following changes:

Please keep the structure and numbering of chapters according to the guidelines of the journal.

Introduction too long. Give the most important examples and explain the need for a new method. And move the details to the discussion.

I would renumber chapters 2-4 as one chapter with subtitles. shorten and insert into the method as a description of the measuring device

Chapter 5 does not have a specific research objective

section 5.1.3 please state the reason why protective equipment cannot be used

5.1.4. explain why OptiTrack uses a filter and FRTMS does not

Figures 7,8,9 I'm not sure if these figures are needed to show or do they provide important information? And they take up a lot of space, one or two examples and a description of the others will suffice

In the discussion at the beginning, emphasize the most important results and the summary of the research. Then compare the method with other systems as it is now in the introduction.

chapter 6 to discussion or shortened

Author Response

Response to Reviewer 3 Comments

On behalf of my co-authors, thank you very much for giving us an opportunity to revise our manuscript, we appreciate reviewers very much for your constructive and positive comments and suggestions which helped us improve our manuscript entitled “Development and Evaluation of a Full-waveform Resistance Training Monitoring System Based on a Linear Position Transducer”. (Manuscript ID: sensors-2156481).

We have studied reviewer’s comments carefully and have made revision which marked in red font in the copy of the revised manuscript. We have tried our best to revise our manuscript according to the comments from reviewers.

Thank you and best regards.

Yours sincerely,

Changda Lu

E-mail: luchangda@bsu.edu.cn

Reviewers' comments:

The presented concept of a new measurement method is an interesting supplement to the existing solutions and is worth considering.

However, the text is very long and quite hard to read. There are too many goals in one article. I would suggest considering whether it would not be worth dividing this text into two or even three reports. The first topic is a review of existing studies and a comparison of methods. The second topic is the presentation of the new device with technical details. And the third topic is the use of the device in pilot studies and comparing the results with other methods. This would greatly simplify the message and the topics would be clearer and easier for the reader.

 However, if you wish to publish it in this form, I suggest the following changes:

Please keep the structure and numbering of chapters according to the guidelines of the journal.

Introduction too long. Give the most important examples and explain the need for a new method. And move the details to the discussion.

I would renumber chapters 2-4 as one chapter with subtitles. shorten and insert into the method as a description of the measuring device.

Chapter 5 does not have a specific research objective.

section 5.1.3 please state the reason why protective equipment cannot be used.

5.1.4. explain why OptiTrack uses a filter and FRTMS does not.

Figures 7,8,9 I'm not sure if these figures are needed to show or do they provide important information? And they take up a lot of space, one or two examples and a description of the others will suffice.

In the discussion at the beginning, emphasize the most important results and the summary of the research. Then compare the method with other systems as it is now in the introduction.

chapter 6 to discussion or shortened.

To reviewer 3

We thank reviewer 3 for the critical comments and helpful suggestions. We have taken all these comments and suggestions into account, and they have improved our manuscript considerably.

Point 1: The text is very long and quite hard to read. There are too many goals in one article. I would suggest considering whether it would not be worth dividing this text into two or even three reports. The first topic is a review of existing studies and a comparison of methods. The second topic is the presentation of the new device with technical details. And the third topic is the use of the device in pilot studies and comparing the results with other methods. This would greatly simplify the message and the topics would be clearer and easier for the reader.

Response 1: Thank you for your suggestion. For the problem that the text is too long and hard to read, we have simplified the examples of the Introduction and refactored the Materials and Methods to improve readability. For the problem that one article has too many goals, we reorganized the structure and content of the manuscript and wished to publish it with the theme of development and validation of a system.

Point 2: Please keep the structure and numbering of chapters according to the guidelines of the journal.

Response 2: Thank you very much for your reminder. With your advice, we have rewritten the article in the format of “introduction, materials and methods, results, discussion and conclusions” according to the journal's guidelines.

Point 3: Introduction too long. Give the most important examples and explain the need for a new method. And move the details to the discussion.

Response 3: Thank you very much for your suggestions. According to your suggestions, we have kept only the most important examples of the Introduction and moved others to Discussion for detailed discussion. Additionally, we also removed the explanation for IMU-, camera-, laser-, and LPT-based systems to simplify the Introduction。

Revised details on manuscript:

The revised introduction section is as follows:

…..

Resistance training (RT) is necessary for the development of muscle strength and is a fundamental component of many athletes' periodized training programs[1,2]. Monitoring variables of RT allows coaches to improve their athletes' training performance and optimize corresponding training plans. The RT's effect depends on the operation of the training variables like the load, volume, and type of exercises[3,4,5].

Traditionally, the configuration of the RT’s load is depended on the percentage of one‐repetition maximum (1RM). Given the daily changes in 1RM, the actual and proposed loads make a mismatch[3,6]. Recent studies have concentrated on the kinematic parameters represented by movement velocity to solve the problem of mismatched training loads [7,8]. The movement velocity could estimate the 1RM from the velocity recorded against submaximal load based on load-velocity (L-V) relationships[9,10], prescribe the training loads and volumes accurately and objectively based on the magnitude of velocity loss [11,12], and increase motivation by providing real-time velocity feedback[13,14,15,16]. Furthermore, the muscular and functional performance improved with the small incremental changes in the velocity relative to some reference loads in well-trained athletes[7,17,18,19]. Thus, it is essential to measure movement velocity with an accurate and reliable monitoring system.

In previous studies, the three-dimensional (3D) motion capture system was widely considered a “gold standard” instrument for measuring movement velocity.[20,21,22,23].   However, the 3D motion capture system was not practical for daily training and test scenarios. Therefore, the systems based on the inertial measurement unit (IMU)[24,25], camera[26], laser rangefinder[27] and linear position transducer(LPT)[28,29,30] werewidely used in those scenarios because of their convenience and low cost[31,32,33]. Among the above systems, the LPT-based system was considered as the most valid and reliable measurement system [29,34,35,36].

Previous validity and reliability researches about these systems mostly concentrated on the concentric phase of the movement[29,31,37,38,39,40]. These studies usually monitored the statistical indicators represented by the mean or peak value of velocity during training[41], but these indicators for the whole movement may ignore the full waveform data of the movement[42]. Studies have shown that the movements were characterized by a combination of concentric and eccentric muscle action in almost any type of sport[43]. Moreover, the eccentric phase of the training movements could optimize improvements to maximal muscular strength, power development and prevent sports injuries[44,45,46,47]. Menrad et al. used the MV of concentric and eccentric phases on Bland-Altman-diagrams and linear regression to test the accuracy of the three systems during three exercises. LPT and IMU provided valid results determining MV in back squat, deadlift and barbell rowing, but PUSH 2.0 provided not fully valid data[48]. Concentric and eccentric phase studies used the statistical indicators of the velocity to evaluate the validity of the system. Unfortunately, none of these studies focused on the validity of the full-waveform data during the movement.

To monitor the training process more comprehensively and provide valid data for later research on the refinement of training, this study has developed the full-waveform resistance training monitoring system (FRTMS) to acquire the full-waveform data and verified its validity of measuring velocity during the whole movement. Additionally, this study also applied the FRTMS to a real training scenario to explore its effects on training.

…...

Point 4: I would renumber chapters 2-4 as one chapter with subtitles. shorten and insert into the method as a description of the measuring device.

Response 4: Thank you very much for your comments. According to your comments, we reorganized chapters 2-4 as one chapter and inserted them into chapter 2. Materials and Methods. In addition, we reorganized the description of the FRTMS and redescribed it in two parts: Data Acquisition Device and Software Platform.

Revised details on manuscript:

The structure of the revised manuscript is as follows:

…..

  1. Introduction
  2. Materials and Methods

       2.1 Full-waveform Resistance Training Monitoring System

              2.1.1 Data Acquisition Device

              2.1.2 Software Platform

2.2 Validity of the Full-waveform Resistance Training Monitoring System

2.2.1 Subjects

2.2.2 Equipment Setup

2.2.3 Experimental Procedure

2.2.4 Statistical Analysis

2.2.5 Data Processing and Analysis

  1. Results
  2. Discussion
  3. Conclusion

…..

Point 5: Chapter 5 does not have a specific research objective.

Response 5: Thank you very much for your reminder. After your reminder, we have added the research objective in the revised chapter 2.2.

Revised details on manuscript

The revised details of the manuscript are in chapter 2.2:

……To access the validity of the FRTMS, an experiment was designed and performed. ……

Point 6: Section 5.1.3 please state the reason why protective equipment cannot be used.

Response 6: Thank you very much for your suggestion. The explain of this question is as follows: Studies have shown that the protective equipments could improve movement velocity. That means the subjects with protective equipments will have faster movement during squat exercise than subjects without those. Wearing protective equipments can affect the measurement of velocity in the experiments. To ensure consistency of measurement, all subjects were not allowed to use protective equipment during the squat. All subjects were protected by trainers during the squat to prevent sports injuries.

Now we have summarized the explanation in the revised manuscript.

References
These data suggest that the use of a weight belt during the squat exercise may affect the path of the barbell and speed of the lift without altering myoelectric activity. This suggests that the use of a weight belt may improve a lifter's explosive power by increasing the speed of the movement without compromising the joint range of motion or overall lifting technique.

Zink, A.J.; Whiting, W.C.; Vincent, W.J.; MCLAINE, A.J. The effects of a weight belt on trunk and leg muscle activity and joint kinematics during the squat exercise. The Journal of Strength & Conditioning Research 2001, 15, 235-240. doi: https://doi.org/10.1519/00124278-200105000-00013

Revised details on manuscript

The revised details of the manuscript are in chapter 2.2.3 Experimental Procedure:

….Since studies have shown that the protective equipments could improve movement velocity[49], subjects were prohibited from wearing any protective equipment (wrist guard, belt, and knee strap) during the examination to ensure uniformity and accuracy of measurements.….

Point 7: 5.1.4. explain why OptiTrack uses a filter and FRTMS does not.

Response 7: Thank you for your question. The explain of this question is as follows: Filtering can improve data quality. The OptiTrack system were filtered by a 6 Hz low-pass fourth-order butter worth filter to remove any high-frequency noise. However, for a more realistic assessment of the accuracy of the system, we used the FRTMS’s raw data without filtering. Additionally, in future research, we plan to carry further out studies on the effect of different filtering methods on FRTMS’s data.

References
Marker track reconstruction and automated marking were initially performed with Motive (version 2.2.0, OptiTrack Inc., Corvallis, OR, USA) for the marker data collected by the OptiTrack. Each trial was then visually inspected, and the unmarked trajectories were manually labeled and exported as a .trc file.The data were then filtered by a 4th-order Butterworth low pass filter (6 Hz) to remove any high-frequency noise before being imported into a 42-DOF skeletal model in OpenSim.

Wu, Y.; Tao, K.; Chen, Q.; Tian, Y.; Sun, L. A Comprehensive Analysis of the Validity and Reliability of the Perception Neuron Studio for Upper-Body Motion Capture. Sensors 2022, 22, 6954. https://doi.org/10.3390/s22186954

Revised details on manuscript

The revised details of the manuscript are in chapter 2.2.4 Data Processing and Analysis:

….. Furthermore, the data of the OptiTrack system were filtered by a 6 Hz low-pass fourth-order butter worth filter to remove any high-frequency noise. For a more realistic assessment of the accuracy of the device’s raw data, the measurements of the FRTMS were used unfiltered. .….

Point 8: Figures 7,8,9 I'm not sure if these figures are needed to show or do they provide important information? And they take up a lot of space, one or two examples and a description of the others will suffice.

Response 8: Thank you very much for your suggestions. Figure 7,8,9 shown the regression analysis for the concentric phase data, eccentric phase data and full-waveform data of the system respectively to display the validity of the system during different phases. The subplots in each graph represent the validity of the system under different relative loads(30%-90%1RM).

Point 9: In the discussion at the beginning, emphasize the most important results and the summary of the research. Then compare the method with other systems as it is now in the introduction.

Response 9: Thank you very much for your suggestions. According to your suggestions, we have emphasized the most important results and the summary of the research at the beginning of the Discussion. A Comprehensive Analysis of the Validity and Reliability of the Perception Neuron Studio for Upper-Body Motion Capture, we have compared the method of the validition with other three studies.

Revised details on manuscript

The revised details of the manuscript are in chapter 4:

…..

In this study, we proposed and validated the FRTMS as a whole movement process monitoring solution to acquire and analyze the full-waveform data of resistance training.

Despite the previous studies verifying the validity of different systems from different relative loads by measuring various statistical metrics, none of these studies focused on the validity of the data during the whole movement. Ferro et al. used the concentric phase’s Peak Velocity (PV) for evaluated the typical error (TE) and the ICCs to verify that the LPT, IMU and force platform could provide reliable measurements during the countermovement jump exercise. McGrath et al. used the concentric phase’s mean velocity (MV) for linear regression and evaluated the intraclass correlation coefficients (ICC) to verify that the LPT could provide valid data during the bench press[50]. Thompson et al. used the concentric phase’s MV and PV for evaluated the TEs and coefficient of variations (CVs) to verify that the LPT provided the most valid and reliable measurements for the back squat and power clean, followed by camera-based system and IMU for the back squat[40].

Compared with these studies, our study supplemented the CMC as a validity indicator to evaluate the level of concordance of the full waveform data acquired by various systems.

…..

Point 10: Chapter 6 to discussion or shortened.

Response 10: Thank you very much for your suggestions. We have insert the application study into Discussion.

Revised details on manuscript

The revised details of the manuscript are in chapter 4:

…..

The explanation of the FRTMS underestimated the velocity may be associated with the different resolution accuracy for the displacement of FRTMS and OptiTrack systems. The LPT has a slightly lower resolution of displacement than the motion capture system. Thus, the OptiTrack system can capture small displacement and velocity changes before the start and after the end of the motion, whereas the FRTMS cannot. Thus, the OptiTrack system will incorporate these small velocity values into the MV calculation, making its MV larger than that of FRTMS.

For further testing the application of the system in practical training, FRTMS was also used in an intervention study that compared the effects of a 10% velocity loss threshold VBT intervention and a traditional percentage-based training (PBT) intervention on lower limbs explosive power (Figure 10). The study consisted of three times 1RM tests (pre-test, mid-test, and post-test) and a six-week, twice-weekly back squat intervention. Before the intervention study, all participants who met the criteria of the experiment were informed of the procedures and signed a written informed consent form. The study protocol adhered to the principles of the Helsinki Declaration and was approved by the institutional review board. During the 1RM test, the LPT of the FRTMS was attached to the barbell to monitor the MCV of each squat.

Preventing sports injury: the participants avoid attempting heavier loads when their MCV was below the squat minimum velocity threshold (0.3 m/s)[51].

Monitoring movement quality: the FRTMS also monitored the depth of each squat, ensuring that all repetitions were performed at a qualified depth.

During the intervention, the system monitored the MCV, position, and instantaneous velocity of the VBT group.

Providing velocity feedback: Real-time velocity feedback improved theparticipants' training motivation and assisted the coach to perform the intervention.

Adjusting training load: comparing today's reference velocity (the MCV of the first squat during 80% of RM) with the corresponding velocity at the 1RM test, if today's reference velocity was 0.06 m/s lower or higher than the velocity at the 1RM test, the training load needs to adjust by ± 5% of the tested 1RM[12].

During the data processing, the full-waveform velocity of three 1RM tests, as well as the MCV and the maximum concentric velocity of each squat, during the six-week intervention provided by the monitoring system were used in the statistical process.

Monitoring the enhancement of movement: comparing the duration of the stricking region and the depth of squat before and after the intervention, the full-waveform velocity can indicate the enhancement of the squat action.

Monitoring the detailed improvement of the training performance: comparing the change of the MCV, the maximal concentric velocity and the time to peak velocity during the intervention can reflect the detailed improvement of the training performance. For example, after a training intervention, the 1RM value of the athletes did not change. However, their time to peak velocity decreased by 50 milliseconds or the duration of the sticking region decreased by 50 milliseconds, which for athletes could be of major benefit to the explosive power of the lower limbs[42].

This study tested the validity of FRTMS during squats performed on the Smith Machine. However, the daily training also included other multi-directional free weight exercises different from the limited movement patterns used in research. Additionally, the subjects of this study did not represent all users of this system. Based on the above limitations, future studies would recruit more subjects to examine the validity of FRTMS during more free weight exercises, such as snatch, power clean and jerk.

…..

Round 2

Reviewer 2 Report

The current manuscript is revised appropriately.

Reviewer 3 Report

Dear authors

Thank you very much for the clarifications and re-editing the text as suggested. The result is acceptable.